# Compressed fluorescence lifetime imaging via combined TV-based and deep priors

**Chao Ji[1,2], Xing Wang[1,2], Kai He[1], Yanhua Xue[1], Yahui Li[1], Liwei Xin[1], Wei Zhao[1,2,3], Jinshou Tian [1,2]*, Liang Sheng[4]***

1 Key Laboratory of Ultra-fast Photoelectric Diagnostics Technology, Xi'an Institute of Optics and Precision Mechanics (XIOPM), Chinese Academy of Sciences (CAS), Xi'an, Shaanxi, China, 2 University of Chinese Academy of Sciences, Beijing, China, 3 State Key Laboratory of Transient Optics and Photonics, Xi'an Institute of Optics and Precision Mechanics, Chinese Academy of Sciences, Xi'an, Shaanxi, China, 4 The Northwest Institute of Nuclear Technology, Xi'an, China

* tianjs@opt.ac.cn (JT); shengliang@tsinghua.org.cn (LS)

**Data Availability Statement:** All relevant data are within the manuscript and its Supporting Information files

**Funding:** The authors received no specific funding for this work.

## Abstract

Compressed fluorescence lifetime imaging (Compressed-FLIM) is a novel Snapshot compressive imaging (SCI) method for single-shot widefield FLIM. This approach has the advantages of high temporal resolution and deep frame sequences, allowing for the analysis of FLIM signals that follow complex decay models. However, the precision of Compressed-FLIM is limited by reconstruction algorithms. To improve the reconstruction accuracy of Compressed-FLIM in dealing with large-scale FLIM problem, we developed a more effective combined prior model $3DTG_pV\_net$, based on the Plug and Play (PnP) framework. Extensive numerical simulations indicate the proposed method eliminates reconstruction artifacts caused by the Deep denoiser networks. Moreover, it improves the reconstructed accuracy by around 4dB (peak signal-to-noise ratio; PSNR) over the state-of-the-art TV+FFDNet in test data sets. We conducted the single-shot FLIM experiment with different Rhodamine reagents and the results show that in practice, the proposed algorithm has promising reconstruction performance and more negligible lifetime bias.

## 1. Introduction

Widefield fluorescence lifetime imaging (FLIM) is widely used in biomedical diagnostics and flow quantitative measurements, such as cancer diagnosis and treatment monitoring [1, 2], identifying species concentration from reactive-flow systems [3], and understanding the transient evolutionary behavior of eddies in highly turbulent flames [4]. Most of these examples are non-repeatable transient events that demand a single-shot widefield measurement method. However, performing high precision widefield lifetime measurements and quantitative analyses have always been a significant challenge in this field.

The traditional widefield FLIM approaches, including time-correlated single-photon counting (TCSPC) [5, 6], streak camera [7], and single-photon avalanche diode (SPAD) [8, 9] possess high temporal resolution. Nevertheless, they require repeated measurements to obtain the widefield fluorescence lifetime. Recently, a snapshot compressive imaging (SCI) method,

**Competing interests:** The authors have declared that no competing interests exist.

compressed ultrafast photography (CUP), has emerged as a potential solution for snapshot widefield FLIM [10]. Compared to traditional methods, CUP is the only passive 2D technology with picosecond to femtosecond time resolution, which can acquire complete 2D transient processes within a snapshot.

The CUP system is a combination of the streak camera and compressive sensing methods. The typical CUP process is to map 3D encoded data onto a 2D detection array, and then restore the original information through compressed sensing algorithms. However, the data reconstruction step of CUP is a complex task. Significantly, the reconstruction quality of the image deteriorates rapidly with increasing sequence depth. To solve this issue, numerous algorithms have been designed through the exploration of underlying sparsity structures. Plug and Play(PnP) [11] is a typical SCI framework that allows the matching of flexible state-of-the-art forward models with advanced priors or denoising models. On this basis, GAP-TV has become a popular low memory and fast SCI algorithm that combines generalized alternating projection (GAP) and Total Variation (TV) [12]. Denoiser based on block similarity such as block-matching and 3D filtering (BM3D) [13] and weighted nuclear norm minimization (WNNM) [14] enjoy more effective sparsity representation than TV. However, these methods have high computational complexity and often take several hours, while the TV algorithm only takes a few minutes. As a result, BM3D and WNNM are rarely used in Compressed-FLIM, especially when real-time imaging is required.

In contrast to conventional denoisers, Deep denoiser networks such as FFDNet [15] and FastDVDnet [16, 17] resolve the common sparsity representation problem in local similarity and motion compensation while enjoying fast computing speed. However, due to limited priors with the training sets, Deep denoiser networks are required to extract artifacts in the reconstruction process, leading to confusing results. To take advantage of both the Deep denoiser network and TV model, H. Qiu et al. proposed a combined denoiser TV+FFDNet and achieved superior performance to previous algorithms [18].

Inspired by combined priors, we further explore a more effective combination of traditional denoisers and Deep denoiser networks. In this paper, we devise a $3DTG_pV$ denoiser by exploring the underlying sparsity of signals in space-time and the superiority of the non-convex $\ell_p(0 < p < 1)$ norm in minimizing convergence. Meanwhile, by further combining the video denoising network FastDVDnet, we develop a novel combination prior, named $3DTG_pV\_net$.

We make various simulations based on the CUP framework and determine that the proposed $3DTG_pV\_net$ prior offers a ~4dB improvement in peak signal-to-noise ratio (PSNR) compared with the TV+FFDNet prior in runner test sets. Meanwhile, the reconstruction artifacts caused by Deep denoiser networks are successfully eliminated. Besides, we conduct a widefield Compressed-FLIM experiment and obtain 70 consecutive high-resolution images within a single snapshot. Compared with the lifetime bias of the reconstructed data with TV+FFDNet, our method provides higher lifetime evaluation accuracy.

## 2. Principle of compressed-FLIM

A schematic diagram of the compressed ultrafast photography-FLIM (Compressed-FLIM) is illustrated in Fig 1. It comprises three parts: generation of widefield fluorescence signals, data acquisition, and data reconstruction. Unlike the previous scheme [10], we use a transmissive mask rather than the reflective a digital mirror device (DMD) as the spatial encoder.

### 2.1 Generation of widefield fluorescence signals

A 515nm femtosecond laser (200fs) beam passes a cylindrical lens into a laser sheet. The laser sheet illuminates a Rhodamine water solution. Behind the Rhodamine solution, a 515nm filter

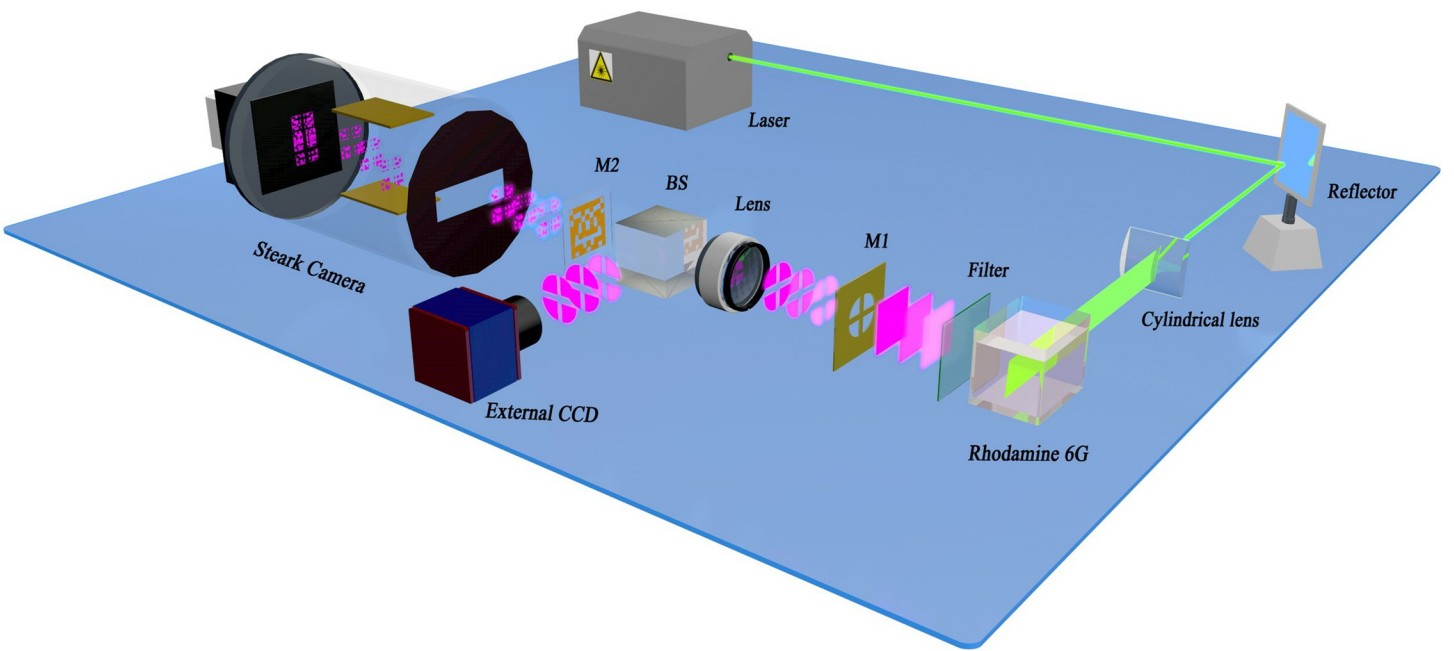

**Fig 1. Schematic diagram of CUP-FLIM.** M1: a pre-designed circular mask with a central cross; M2: the fixed binary mask; BS: beam splitter.

is positioned to filter excitation light. The fluorescence signals pass through a pre-designed circular mask (M1) cut with a cross to highlight spatial recognition, generating shaped fluorescence signals. The diameter of M1 is 35mm.

## 2.2 Data acquisition

After passing through a lens, the shaped signals are divided into two beams by a beam splitter (BS). One sub-signal is directly detected with an external charge-coupled device (CCD) image sensor (Hamamatsu C11440). The other is spatially encoded through a binary mask M2 and recorded by a Streak Camera (XIOPM 5200). The layout of M2 is a random pattern with a pixel resolution of $250 \times 250$, and the size of a single-pixel is $20 \times 20\mu m$. To ensure entire imaging of the targets, the slit of the Streak Camera is fully open (~5 mm), and the image plane of the Streak Camera is adjusted at M2.

In the acquisition section of Streak Camera, the encoded signal undergoes photoelectric conversion at the cathode, then the electric signals at different times are deflected by the slope voltage to various positions on the fluorescent screen. Finally, photons are emitted and collected by the internal CCD ($512 \times 512$ binned pixels; $4 \times 4$ binning). The size of the binned pixels is $26 \times 26 \mu m$.

A DMD is the typical encoder in the CUP system. However, for weak fluorescence acquisition, the fixed binary mask [19] significantly improves the signal-to-noise ratio (SNR) by its transmission characteristics. We randomly generated multiple groups of coding layouts through MATLAB, and selected the best coding layout through simulation results. The signals transmittance rate is ultimately set to 25%.

## 2.3 Data reconstruction

The fluorescence signals can be regarded as a data cube $I(x, y, t)$. In the external CCD view, the cube is directly integrated along the time direction, and the measured data from the CCD can

be expressed as $E_c = \int I(x, y, t)dt$. From the perspective of Streak Camera, operator $T$ carries out spatial coding of the cube, and operator $S$ executes the shearing of signals from the coding cube to the tilted coding cube. Ultimately, the accumulation of tilted coding cubes along the time direction is represented by operator $C$. The entire data acquisition process in Streak Camera view can be described as $E_s = TSCI(x, y, t)$.

Data reconstruction is an ill-condition inverse process. Adding sparsity constraints to the least-squares method realizes the stable reconstruction of the algorithm. The optimization problem of CUP-FLIM can be expressed as:

$$\underset{I}{\mathrm{argmin}} \parallel TSCI(x, y, t) - E_s \parallel_2^2 + \mu \parallel \int I(x, y, t)dt - E_c \parallel_2^2 + \lambda\varphi(I) \tag{1}$$

where the first and the second terms are fidelity terms with data collected by the Streak Camera and external CCD, respectively. The last term $\varphi(I)$ represents the prior used to impose sparsity features to signals while $\mu$ and $\lambda$ are weight parameters. In the next section, we will describe the implementation process of the proposed algorithm and the innovative sophisticated prior.

## 3. Reconstruction algorithm

### 3.1 $3DTG_pV$ priors

Prior plays a key role in the reconstruction algorithms of compressed sensing. The $\ell_0$ norm prior is the sparsest representation, as it counts the number of nonzero entries in signals. However, it is extremely challenging to process numerically. For solving the dilemma of algorithms without convergence, Donoho. et al. verified the approximate equivalence of the $\ell_1$ and $\ell_0$ norms [20].

Formally, the $\ell_1$ norm minimization can be expressed as

$$\underset{u}{\mathrm{argmin}} \parallel Au - b \parallel_2^2 + \lambda\|u\|_1 \tag{2}$$

In the research area of images, by considering the spatial smoothing properties of natural signals, the generalized form of $\ell_1$ norm total variation (TV) has been proved to be far sparser when applying the minimization principle of image gradient shown in Fig 2A. The TV prior obeys

$$\Phi_{TV}(u) = \sum_{i,j,t} \sqrt[2]{\left(|u_{i,j,t} - u_{i-1,j,t}|^2 + |u_{i,j,t} - u_{i,j-1,t}|^2\right)}. \tag{3}$$

Next, we will briefly introduce three generalized forms (3DTV, TGV, $T_pV$) based on the TV prior that strengthen sparsity representation.

We define

$$D_i = |u_{i,j,t} - u_{i-1,j,t}|, D_j = |u_{i,j,t} - u_{i,j-1,t}|, D_t = |u_{i,j,t} - u_{i,j,t-1}|. \tag{4}$$

$$\text{(a)} \qquad\qquad\qquad \text{(b)} \qquad\qquad\qquad \text{(c)}$$

**3.1.1 Stretch in the spatial domain–TGV.** Total generalized variation (TGV) is a second-order gradient minimization proposed by Kunisch. et al. [21]. It incorporates more adjacent elements than TV and regards the second-order gradient of images as the sparse coefficients, as Fig 2B illustrates. For mathematical imaging problems, TGV is an effective approach that enhances the details of high-frequency signals and eliminates staircasing effects [22]. It can be

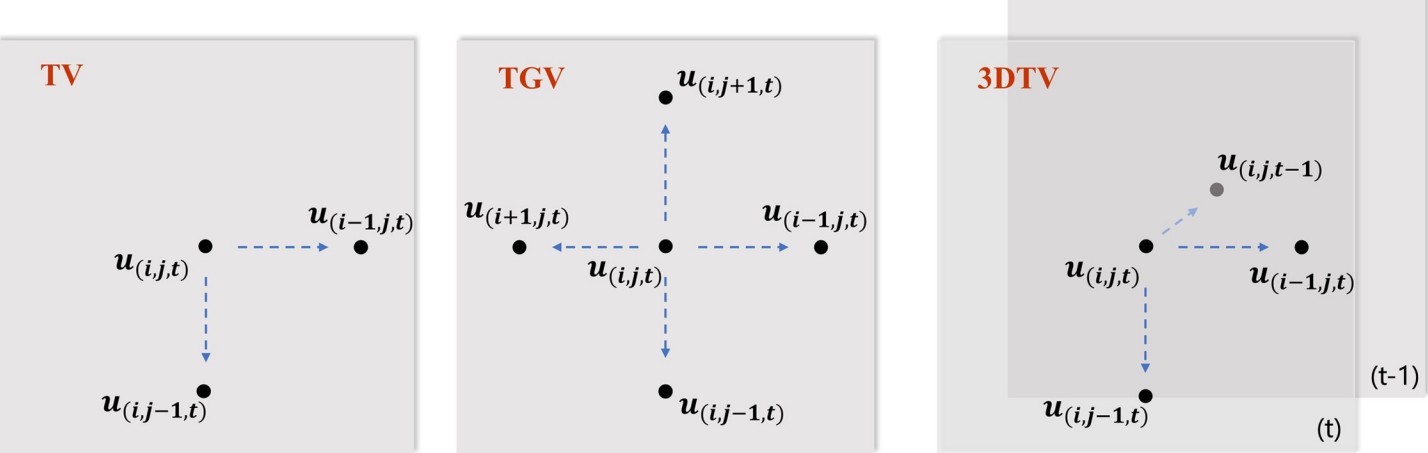

**Fig 2.** Associated elements among the different priors (a)TV (b) TGV (c) 3DTV.

expressed as

$$\Phi_G(u) = \sum_{i,j,t} \sqrt[2]{\left(D_{i+1} - D_i\right)^2 + \left(D_{j+1} - D_j\right)^2}. \tag{5}$$

**3.1.2 Stretch in the time domain– 3DTV.** The TV prior merely considers image similarity in continuous 2D space but ignores the similarity of adjacent elements [23] in the time direction. 3DTV introduces the 3D sparsity constraint of fluorescence signals shown in Fig 2C, and can be represented as

$$\Phi_{3D}(u) = \sum_{i,j,t} \sqrt[2]{\left(D_i^2 + D_j^2 + D_t^2\right)}. \tag{6}$$

Furthermore, given that the time-domain correlation decreases with the increase of motion scale, we add a time-domain weight parameter $\tau$ $(0 \leq \tau \leq 1)$ to flexibly balance the relevancy between motion scale and time-domain correlation. Therefore, Eq (6) can be rewritten as

$$\Phi_{3D}(u) = \sum_{i,j,t} \sqrt[2]{\left(D_i^2 + D_j^2 + \tau D_t^2\right)}. \tag{7}$$

**3.1.3 $\ell_p$ (0 <p < 1) norm of gradient– $T_pV$.** The $\ell_p$ (0 <p < 1) norm is defined as $\left(\sum_{i=1}^{n} |x_i|^p\right)^{1/p}$, which is closer to $\ell_0$ norm than $\ell_1$ norm in mathematical form, thus approaching the sparsest solution. Our previous work has proved that non-convex optimization algorithm based on $\ell_p$ norm has more vital sparsity constraints even when it belongs to non-convex optimization problem [24]. With superior sparsity performance, the $T_pV$ prior eliminates artifacts and achieves superior reconstruction results [25, 26]. The $T_pV$ prior is expressed as

$$\Phi_p(u) = \sum_{i,j,t} \sqrt[p]{\left(D_i^p + D_j^p\right)} \tag{8}$$

By combining the different merits of the three TV-based priors, we proposed the following $3DTG_pV$ prior:

$$\Psi(u) = \sum_{i,j,t} \sqrt[p]{\left( D_{i+1} - D_i \right)^p + \left( D_{j+1} - D_j \right)^p + \tau \left( D_{t+1} - D_t \right)^p}. \tag{9}$$

### 3.2 PnP- $3DTG_pV\_net$ algorithms

In this section, we propose a novel algorithm based on the PnP-framework $3DTG_p$ V_net prior by combining $3DTG_p$ V prior and Deep denoiser network.

We will introduce the overall algorithm flow presented in Fig 3 and Algorithm 1. The sparse signals of interest $u$ are rebuilt by determining the minimum solution from the following constrained formula

$$(u^{(t)}, v^{(t)}) = \arg\min_{u,v} \frac{1}{2} \| u - v \|_2^2 + \lambda\Psi(v), \tag{10}$$
$$\text{subject to } A_s u = b_s, \ A_c u = b_c$$

where $b_s$ denotes data measured by Streak Camera. $b_c$ signifies data measured by the external CCD, $A_s$ and $A_s$ represent the corresponding projection matrices, respectively, $v$ is an auxiliary variable, and $\lambda$ is an added weight.

According to the generalized alternating projection (GAP) algorithm [12], we update the fidelity and prior term separately. Fig 3 displays the workflow of the PnP-3D$TG_p$ V_net algorithm. For each iteration stage, we first apply the Euclidean projection for updating $u^{(t)}$:

$$u^{(t)} = v^{(t-1)} + \frac{\left[ A_s^\top (A_s A_s^\top)^{-1}(b_s - A_s v^{(t-1)}) + \mu A_c^\top (A_c A_c^\top)^{-1}(b_c - A_c v^{(t-1)}) \right]}{1 + \mu} \tag{11}$$

The update of $v$ is a denoising problem, we execute the denoising process by using 3D$TG_p$V and FastDVDnet [16], respectively

For the 3D$TG_p$V update section

$$v_1^{(t)} = u^{(t)} - \Psi^\top \left( z^{(t)} \right), \tag{12}$$

$$z^{(t)} = \text{clip} \left( z^{(t-1)} + \frac{1}{\alpha} \Psi \left( v^{(t-1)} \right), \frac{\lambda}{2} \right), \tag{13}$$

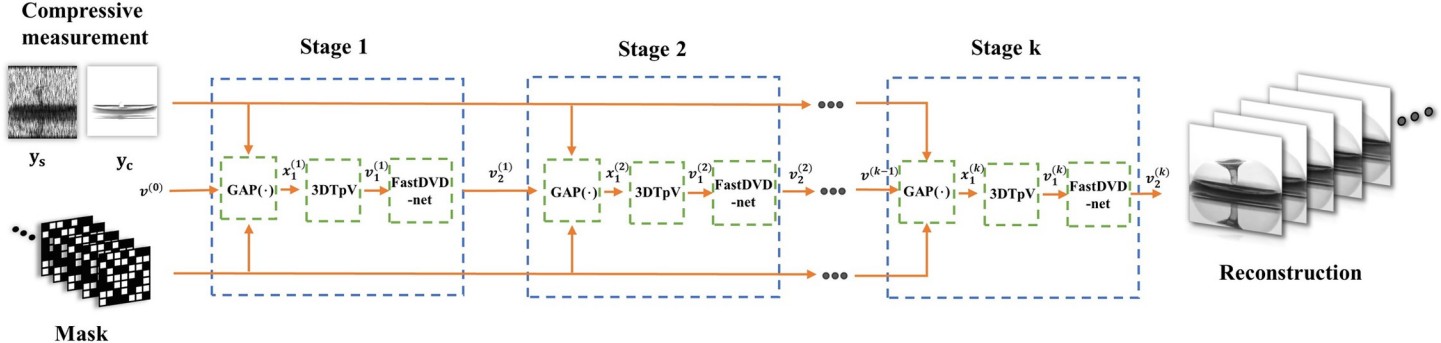

**Fig 3. Workflow of the *PnP- 3DTG_pV_net* algorithm.**

where

$$\text{clip}(\sigma, T) := \begin{cases} \sigma, & \text{if } |\sigma| \le T \\ \text{Tsign}(\sigma), & \text{otherwise} \end{cases} \tag{14}$$

For the Deep denoiser network update section

$$v_2^{(t)} = FastDvDnet\left(v_1^{(t)}\right) \tag{15}$$

**Algorithm 1. PnP- $3DTG_p$ $V\_net$ framework.**

```
Input A_s, A_c, b_s, b_c, Given p ∈ (0,1):

Initialize v_0 = A_s'b_s, μ = 0.1, λ = 0.07

for iteration in range(0, 250):
```

1. Update streak camera's reconstruction data $u_{sc}^{(t)}$ by
$$u_{sc}^{(t)} = \boldsymbol{v}^{(t-1)} + \mathbf{A_s}^\top(\mathbf{A_s}\mathbf{A_s}^\top)^{-1}(\boldsymbol{b_s} - \mathbf{A}\boldsymbol{v}^{(t-1)});$$

2. Update CCD's reconstruction data $u_{CCD}^{(t)}$ by
$$u_{CCD}^{(t)} = \boldsymbol{v}^{(t-1)} + \mathbf{A_c}^\top(\mathbf{A_c}\mathbf{A_c}^\top)^{-1}(\boldsymbol{b_c} - \mathbf{A_c}\boldsymbol{v}^{(t-1)});$$

3. Update $u^t$ by $u^t = \left(x_{sc}^{(t)} + \mu u_{CCD}^{(t)}\right)/(1 + \mu);$

4. $3DTG_pV$ denoising: Update $\boldsymbol{v_1^{(t)}}$ by $\boldsymbol{v_1^{(t)}} = \boldsymbol{u}^{(t)} - \Psi^\top(\boldsymbol{z}^{(t)})$ where
$$\boldsymbol{z}^{(t)} = \text{clip}\left(\boldsymbol{z}^{(t-1)} + \frac{1}{\alpha}\Psi(\boldsymbol{v}^{(t-1)}), \frac{\lambda}{2}\right);$$

5. Deep network denoising: Update $\boldsymbol{v_2^{(t)}}$ by $v_2^{(t)} = FastDvDnet\left(v_1^{(t)}\right);$

6. Update $\boldsymbol{v^{(t)}}$ by $\boldsymbol{v^{(t)}} = \boldsymbol{v_2^{(t)}}$

```
Obtain reconstruction result: u
```

## 4 Simulation results

In the simulation, we compare the reconstruction performances of six priors (TV, $3DTG_pV$, BM3D, TV+FFDNet, TV+FastDVDnet, and $3DTG_p$ V_net) by applying widely-used drop and runner datasets. Each dataset comprises 30 video clips. For initialization, we set $\boldsymbol{v}^{(0)} = A_s^\top b_s$, $z^{(0)} = 0$, $\lambda = 0.07$, $\mu = 0.1$, and $\tau = 0.2$. Each algorithm performs 250 iterations independently. For related Deep network, we directly use the FFDNet model and parameters from https://github.com/cszn/KAIR. Besides, the FastDVDnet model and parameters are from https://github.com/m-tassano/fastdvdnet. The drop and runner datasets are from https://github.com/zsm1211/PnP-SCI/tree/master/dataset/simdata/benchmark.

Figs 4 and 5 present the reconstructed frames (seen in **Visualization 1** and **Visualization 2**) restored with different priors using the two above-mentioned datasets. The 3D$TG_pV$ prior achieves superior detailed than the TV prior. Although BM3D has a more rigorous denoising ability than the TV-based methods, excessive smoothing leads to the loss of image detail. The combined priors based on TV and Deep denoiser networks (TV+FFDNet and TV+FastDVDnet) provide better reconstruction contrast and detail than traditional priors but they expose unsatisfactory artifacts in the reconstructed images. Our proposed combined prior 3D$TG_p$ V_net succeed in eliminating artifacts, leading to more accurate representations of the original images.

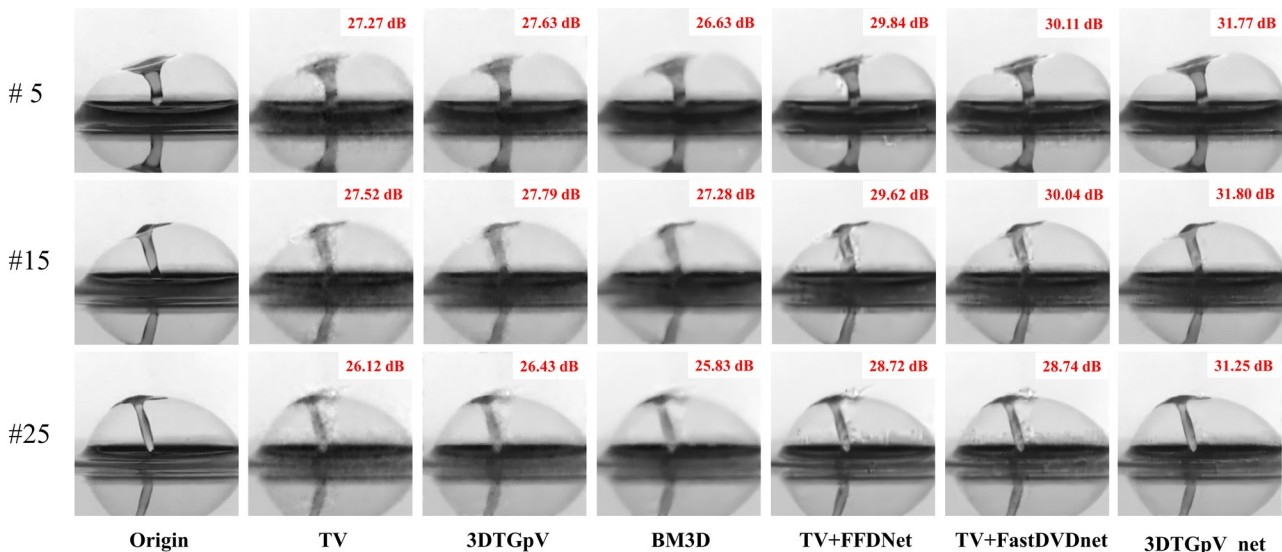

**Fig 4. The 5th, 15th and 25th reconstructed frames based on simulated datasets: Drop.**

We evaluate the quality of the reconstructed images by two indicators: peak signal-to-noise ratio (PSNR) and structural similarity (SSIM).

The PSNR can be calculated by

$$PSNR = 10 \cdot \log_{10} \left( \frac{255^2 \times mn}{\sum_{i=0}^{m-1 n-1} \left[ x(i,j) - y(i,j) \right]^2} \right) \quad (16)$$

where $x$ and $y$ represents the original image and the reconstructed image, respectively. $m$ and $n$ indicates the height and width of the image, respectively.

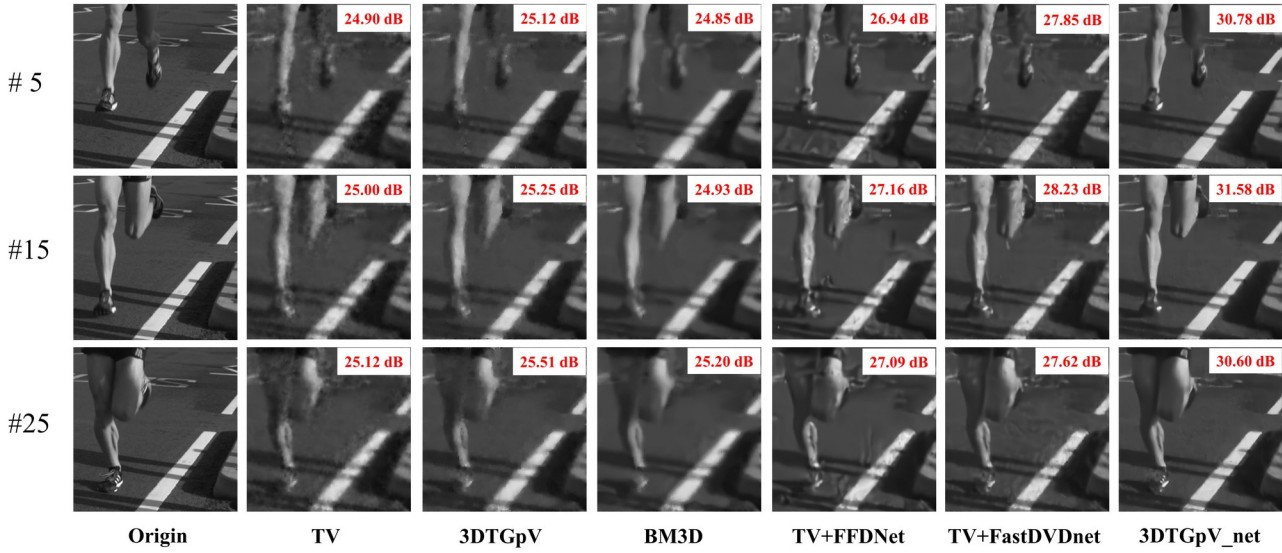

**Fig 5. The 5th, 15th and 25th reconstructed frames based on simulated datasets: Runner.**

**Table 1. Average PSNR and SSIM results.**

| Priors | Drop | | Runner | |
|---|---|---|---|---|
| | PSNR | SSIM | PSNR | SSIM |
| TV | 26.90 | 0.901 | 24.86 | 0.862 |
| 3D$TG_p$V | 27.13 | 0.905 | 25.06 | 0.874 |
| BM3D | 26.57 | 0.891 | 24.81 | 0.862 |
| TV+FFDNet | 29.20 | 0.934 | 26.91 | 0.899 |
| TV+FastDVDnet | 29.76 | 0.942 | 27.85 | 0.914 |
| **3DTG$_p$ V_net** | **31.75** | **0.958** | **30.87** | **0.948** |

The SSIM can be calculated by

$$SSIM(x, y) = \frac{\left(2\mu_x\mu_y + c_1\right)\left(2\sigma_{xy} + c_2\right)}{\left(\mu_x^2 + \mu_y^2 + c_1\right)\left(\sigma_x^2 + \sigma_y^2 + c_2\right)} \tag{17}$$

where $\mu_x$ and $\mu_y$ represents the mean value of the original image and the reconstructed image, respectively. $\sigma_x^2$ and $\sigma_y^2$ are the corresponding variance. $\sigma_{xy}$ means the covariance.

Table 1 presents the average PSNR and SSIM results. We can conclude that the 3D$TG_p$ V_net prior outperforms the other priors in both PSNR and SSIM. Significantly, the proposed prior improve the reconstructed accuracy by approximately 4dB (PSNR) over the state-of-the-art TV+FFDNet in runner test sets.

## 5. Experiments

In the experiments, we record widefield fluorescence data of Rhodamine 6G and Rhodamine B by CUP. The results are reconstructed using both the PnP—TV+FFDNet algorithm and the proposed PnP - 3D$TG_p$ V_net algorithm. We set the CUP time resolution to be 330 *ps*. The reconstruction process is implemented in Ubuntu 20.04 with an NVIDIA GeForce GTX 1650Ti GPU.

Fig 6A presents the streak camera measurement data for Rhodamine 6G. The scanning direction of the data is from top to bottom. Fig 6B and 6C show the widefield fluorescence data rebuilt by the PnP-TV+FFDNet and PnP - 3D$TG_p$ V_net algorithms, respectively. Also, the reconstructed movies are shown in **Visualization 3** and **Visualization 4**. By comparing the two sets of data, it is apparent that our proposed algorithm achieves smoother reconstruction results with fewer artifacts.

In Fig 7A, the measured data of Rhodamine B is displayed. It has a shorter glow duration than Rhodamine 6G. The corresponding rebuild results are shown in Fig 7B and 7C, and the movies are presented in **Visualization 5** and **Visualization 6**. The results indicate that the proposed algorithm achieves better detail reconstruction in various fluorescence environments.

To further analyze the measurement accuracy of widefield fluorescence lifetime, we implement the exponential fitting with the least square method, based on a mono-exponential decay model for each pixel [27].

The measured decay $h(t)$ can be expressed as

$$h(t) = \text{irf}(t) * Aexp\left(-\frac{t}{\tau}\right) + \varepsilon \tag{18}$$

where $A$ represents the amplitude, $\tau$ denotes the lifetime, $\varepsilon$ signifies noise, and irf $(t)$ is the instrument response function (IRF) of the measurement system. Since the full width at half

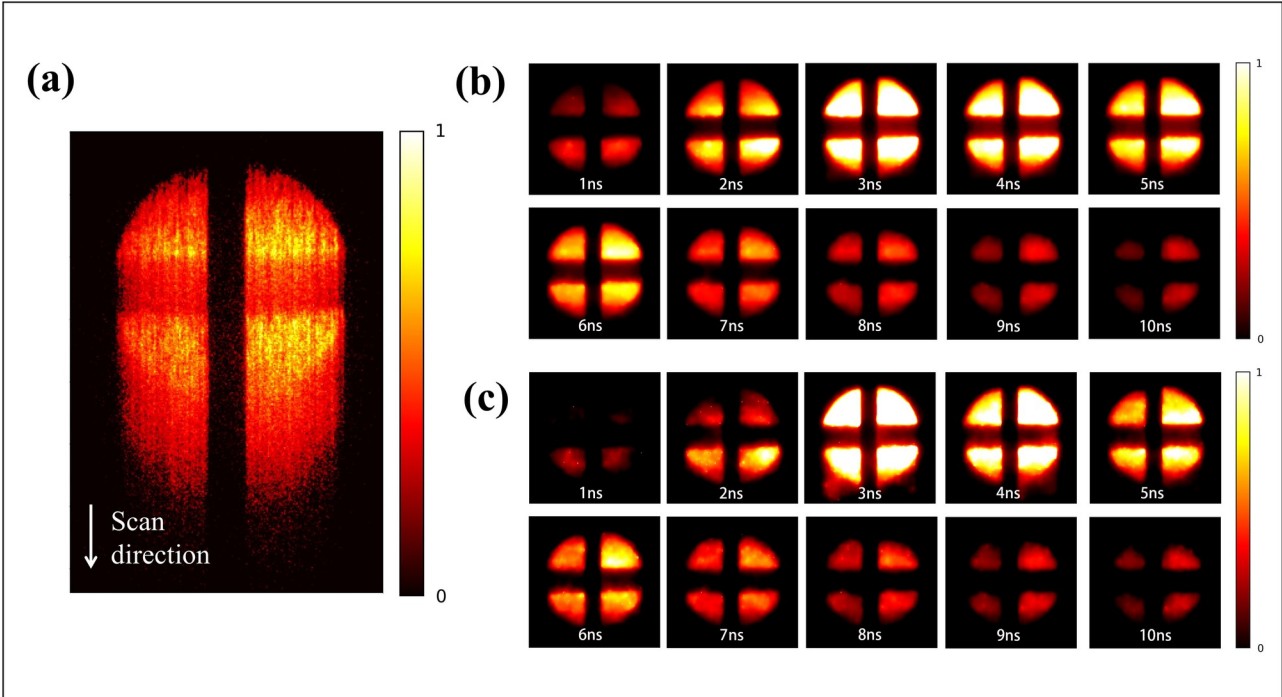

**Fig 6. Measurement and reconstruction data of Rhodamine 6G:** (a) Streak Camera image; (b) Reconstructed frames using PnP-TV+FFDNet algorithm; (c) Reconstructed frames using PnP-3$DTG_p$ V_net algorithm.

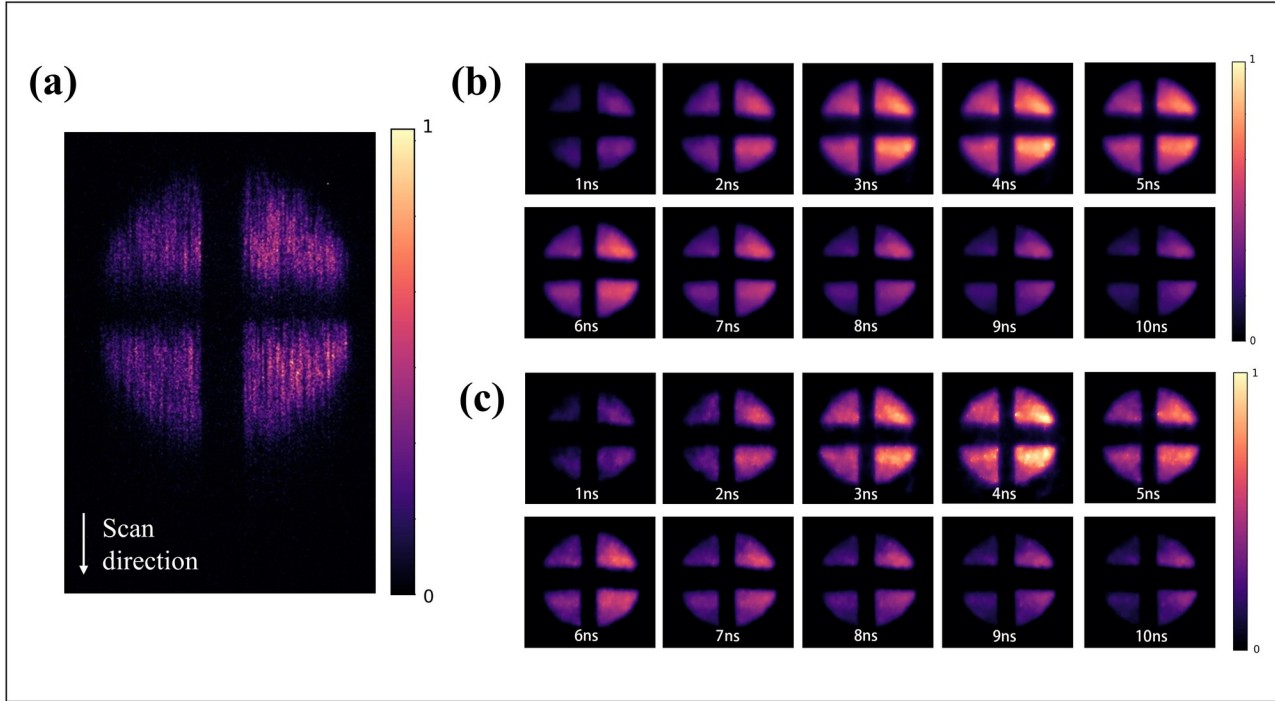

**Fig 7. Measurement and reconstruction data of Rhodamine B:** (a) Streak Camera image; (b) Reconstructed frames using PnP-TV+FFDNet algorithm; (c) Reconstructed frames using PnP-3$DTG_p$ V_net algorithm.

$$3DTG_pV\_net(ours) \qquad TV + FFDnet$$

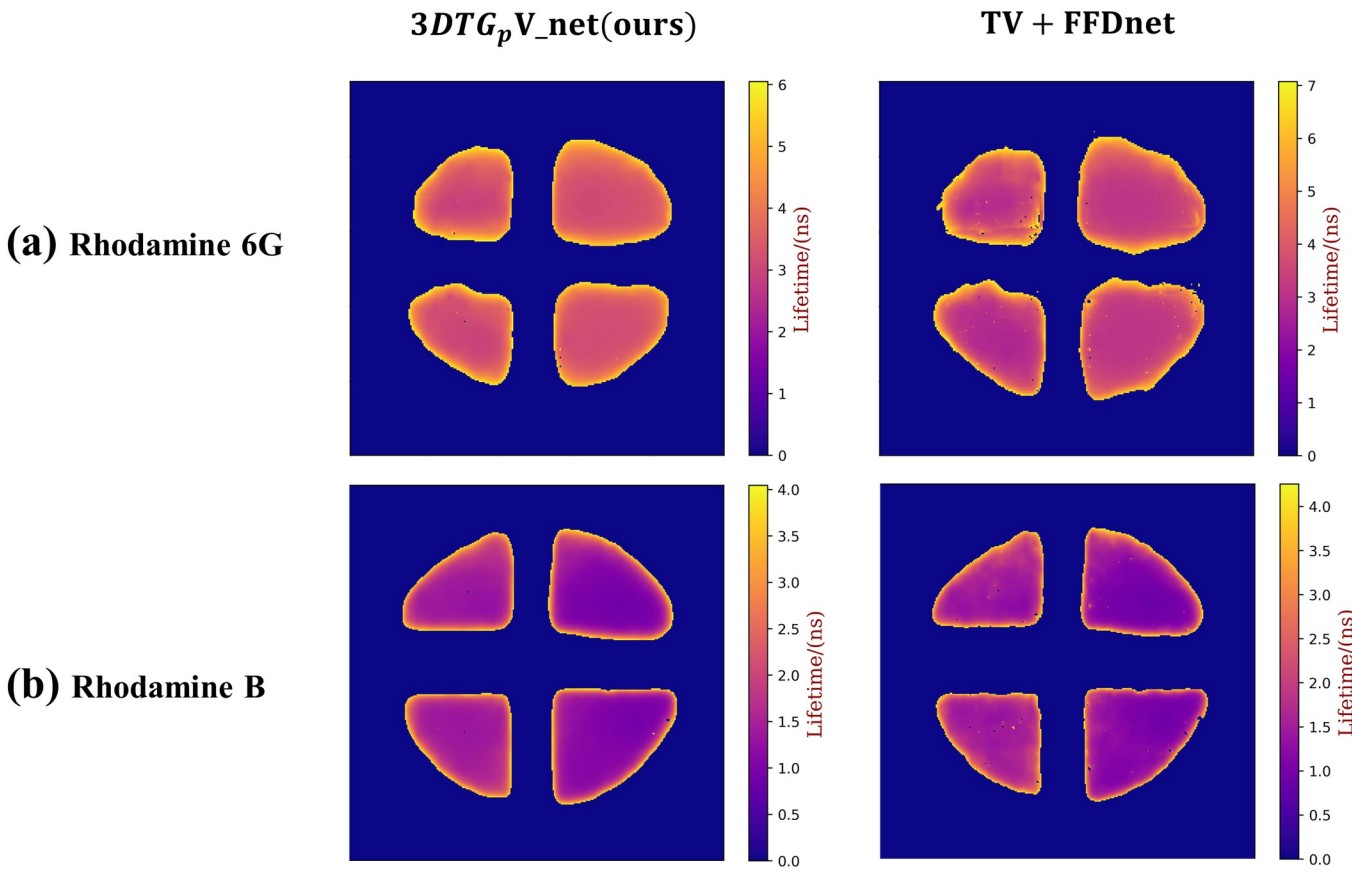

**(a) Rhodamine 6G**

**(b) Rhodamine B**

**Fig 8. Comparison with 2D lifetime images.**

maximum (FWMH) of the laser pulse is 200fs, irf (*t*) can be regarded as a delta function for fluorescence decays with nanosecond lifetimes.

Fig 8A and 8B display the two groups of 2D lifetime images rebuilt using the PnP-TV+-FFDNet and PnP-3$DTG_p$ V_net algorithms, respectively. Besides, Fig 9 shows the reconstruction lifetime bias. For Rhodamine 6G (R6G), the mean lifetime and standard deviation of the proposed algorithm are 3.91 ns and 0.57 ns, respectively. Also, the corresponding values for PnP-TV+FFDNet are 4.41 ns and 1.1 ns. For Rhodamine B (RB), the mean lifetime and standard deviation of the proposed algorithm are 1.68 ns and 0.52 ns, while for PnP-TV+FFDNet they are 1.72 ns and 0.54 ns.

In the slit-scanning mode of the Streak Camera, we re-obtain non-superimposed fluorescence lifetime data as a reference. The single exponential fitting results of Rhodamine 6G and Rhodamine B are 3.62 ns and 1.51 ns, respectively. These improved results demonstrate that our proposed PnP-3$DTG_p$ V_net algorithm produces a bias that is 0.29 ns and 0.17 ns lower than the PnP-TV+FFDNet algorithm.

## 6. Conclusion

In this study, we propose 3DTGp V_net, a highly effective Compressed-FLIM combined prior. Results from numerous simulations and experiments confirm that our proposed method has better reconstruction performance than the existing algorithms and presents higher evaluation accuracy for wide-field FLIM. Besides, this study further confirms that combined priors can

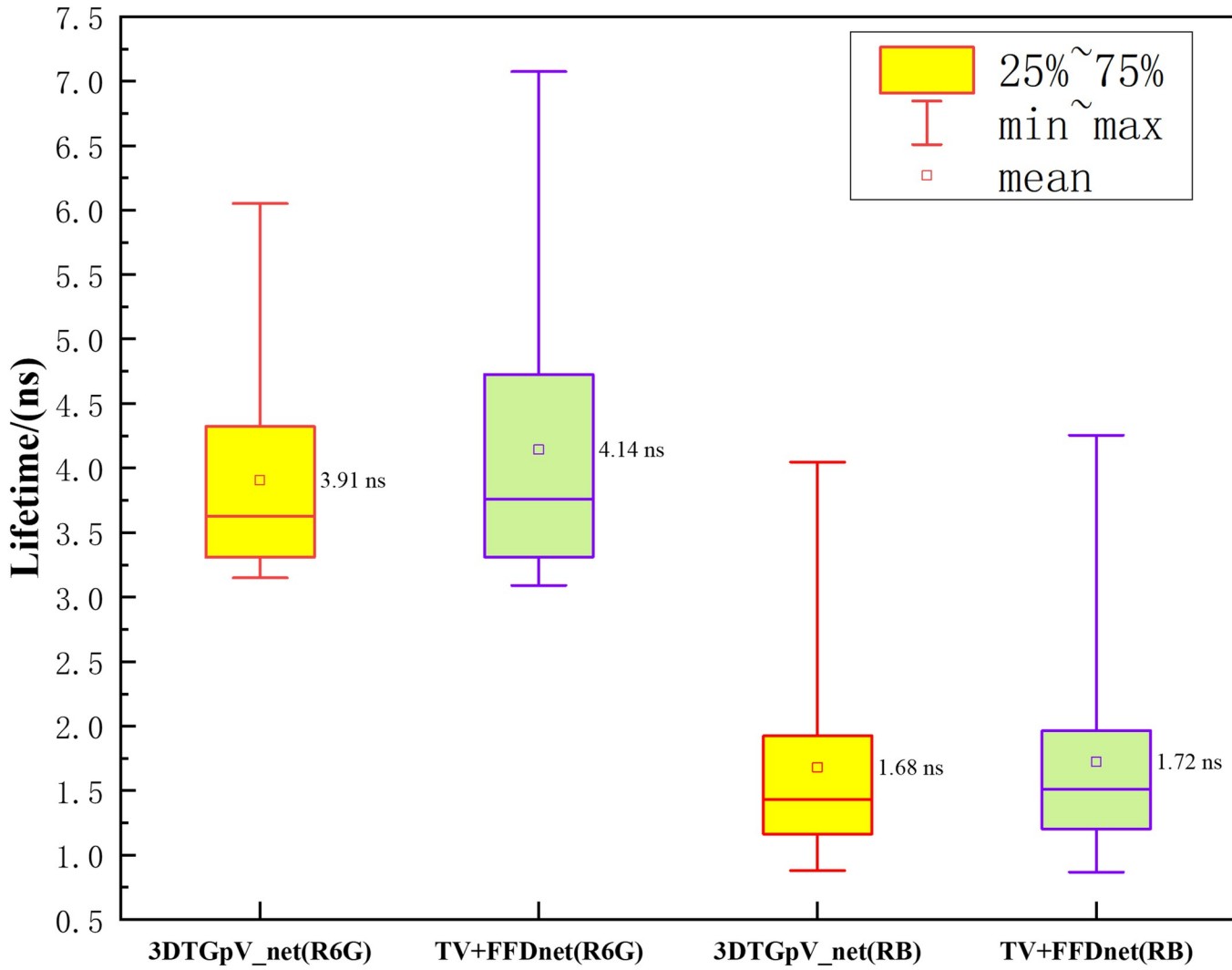

**Fig 9. Comparison with reconstruction lifetime bias.**

effectively complement the advantages of traditional priors and Deep denoiser networks to improve the reconstruction performance of compressive video imaging technology. Lastly, it is noted that our algorithm is a general framework and demanding to relevant SCI systems.

## Supporting information

**S1 Video. Drop.**
(AVI)

**S2 Video. Runner.**
(AVI)

**S3 Video. R6G(3DTGpV_net).**
(AVI)

**S4 Video. R6G(TV+FFDnet).**
(AVI)

**S5 Video. RB(3DTGpV_net).**
(AVI)

**S6 Video. RB(TV+FFDnet).**
(AVI)

## Acknowledgments

The authors would also like to thank Prof. Jinshou Tian and Prof. Liang Sheng for their valuable help and contribution.

## Author Contributions

**Conceptualization:** Chao Ji, Liang Sheng.

**Data curation:** Kai He, Yahui Li, Liang Sheng.

**Formal analysis:** Yahui Li.

**Funding acquisition:** Yanhua Xue, Liwei Xin, Wei Zhao.

**Investigation:** Kai He, Wei Zhao.

**Methodology:** Xing Wang, Liang Sheng.

**Project administration:** Kai He, Wei Zhao.

**Resources:** Liwei Xin, Wei Zhao.

**Software:** Yanhua Xue.

**Supervision:** Yanhua Xue, Liwei Xin.

**Visualization:** Liwei Xin.

**Writing – original draft:** Chao Ji.

**Writing – review & editing:** Xing Wang, Yahui Li, Jinshou Tian, Liang Sheng.

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
