## [Decision Letter · Decision Letter 0]

1 Apr 2022

PONE-D-22-04257Compressed fluorescence lifetime imaging via combined TV-based and Deep PriorsPLOS ONE

Dear Dr. jinshou,

Thank you for submitting your manuscript to PLOS ONE. After careful consideration, we feel that it has merit but does not fully meet PLOS ONE’s publication criteria as it currently stands. Therefore, we invite you to submit a revised version of the manuscript that addresses the points raised during the review process. Please submit your revised manuscript by May 16 2022 11:59PM. If you will need more time than this to complete your revisions, please reply to this message or contact the journal office at plosone@plos.org. Please include the following items when submitting your revised manuscript:A rebuttal letter that responds to each point raised by the academic editor and reviewer(s). You should upload this letter as a separate file labeled 'Response to Reviewers'.A marked-up copy of your manuscript that highlights changes made to the original version. You should upload this as a separate file labeled 'Revised Manuscript with Track Changes'.An unmarked version of your revised paper without tracked changes. You should upload this as a separate file labeled 'Manuscript'.

We look forward to receiving your revised manuscript.

Kind regards,

Li Zeng

Academic Editor

PLOS ONE

Journal Requirements:

2. In the Methods section of your manuscript, please provide full information on the location of the 'drop' and 'runner' datasets. This can take the form of a literature reference, or a URL link to the location of each dataset.

4. Thank you for stating the following in the Funding Section of your manuscript: 

"This work was supported by the Scientific Instrument Developing Project of the Chinese Academy of Sciences (Grant No. GJJSTD20220006), the Youth Program of the National Natural Science Foundation of China (Grant No. 12105360, 62075236), the Science Foundation of the Chinese Academy of Sciences (Grant No. CXJJ-21S006), the Youth Innovation Promotion Association CAS (Grant No. 2020397), and the Rising research star of Shaanxi Province (Grant No. 2021SR5061)."

We note that you have provided funding information that is not currently declared in your Funding Statement. However, funding information should not appear in the Funding section or other areas of your manuscript. We will only publish funding information present in the Funding Statement section of the online submission form. 

"Initials of the authors who received each award"

Reviewers' comments:

Reviewer's Responses to Questions

**Comments to the Author**

1. Is the manuscript technically sound, and do the data support the conclusions?

Reviewer #1: Partly

2. Has the statistical analysis been performed appropriately and rigorously? 

Reviewer #1: Yes

3. Have the authors made all data underlying the findings in their manuscript fully available?

Reviewer #1: No

4. Is the manuscript presented in an intelligible fashion and written in standard English?

Reviewer #1: No

5. Review Comments to the Author

Reviewer #1: This paper by Choa Ji et al describes a compressed sensing technique for fluorescence lifetime imaging. They propose a novel reconstruction technique that uses a novel prior and this is compared to state of the art techniques.

Unfortunately the quality of the writing and some of the methods contained within the paper are not of sufficient standard to fully judge its merits. I therefore recommend a major revision that addresses the points below.

The mathematics is quite sparse, consisting mainly of variations of the prior. I suggest that the authors include text on how this new work fits into the "Plug and play" framework. For publication in PLOS ONE (a journal with a non-specialist readership) I would expect more background information. Please consider if it is possible to recreate the results from the information provided. Provide code in an external repository where possible.

The ground truth for the 2 simulated examples is not shown.

The example experiment contains only Rhodamine 6G are therefore consists of a single lifetime value. The authors should demonstrate that multiple lifetimes within the same scene can be determined. Please add an additional experiment.

There are numerous mistakes in the English writing (too many to list) that make it very hard to understand. This has to be improved before a more thorough review can be performed.

6. PLOS authors have the option to publish the peer review history of their article (what does this mean?). If published, this will include your full peer review and any attached files.

Reviewer #1: No

---

## [Author Response · Author response to Decision Letter 0]

20 May 2022

We would like to express our sincere thanks to the Editor and Reviewer for the constructive and positive comments.

Replies to Editor

Comment 1: The mathematics is quite sparse, consisting mainly of variations of the prior. I suggest that the authors include text on how this new work fits into the "Plug and play" framework. 

Answer: 

We added a text “Algorithm 1. PnP-〖3DTG〗_p V¬_net framework” behind the fig 3 to demonstrate the fitting workflow between the priors and "Plug and play" framework. 

Comment 2: For publication in PLOS ONE (a journal with a non-specialist readership) I would expect more background information. Please consider if it is possible to recreate the results from the information provided. Provide code in an external repository where possible.

Answer:

We added more detailed research background at the beginning of the manuscript introduction. In addition, we have provided all visualization results involving simulations and experiments (seen in Visualization 1-6). As it involves subsequent research, the code will be disclosed later.

Comment 3: The ground truth for the 2 simulated examples is not shown. The example experiment contains only Rhodamine 6G are therefore consists of a single lifetime value. The authors should demonstrate that multiple lifetimes within the same scene can be determined. Please add an additional experiment.

Answer:

We conducted another Compressed-FLIM experiment by using Rhodamine B. Moreover, we confirmed that our algorithm can obtain better reconstruction effect and more accurate lifetime evaluation under different fluorescent samples. 

Comment 4: There are numerous mistakes in the English writing (too many to list) that make it very hard to understand. This has to be improved before a more thorough review can be performed.

Answer:

 Thanks for your suggestion. We have tried our best to polish the language in the revised manuscript. These changes will not influence the content and framework of the paper. And here we did not list the changes but marked in revisions mode in revised paper.

---

## [Decision Letter · Decision Letter 1]

17 Jun 2022

PONE-D-22-04257R1Compressed fluorescence lifetime imaging via combined TV-based and Deep PriorsPLOS ONE

Dear Dr. jinshou,

Thank you for submitting your manuscript to PLOS ONE. After careful consideration, we feel that it has merit but does not fully meet PLOS ONE’s publication criteria as it currently stands. Therefore, we invite you to submit a revised version of the manuscript that addresses the points raised during the review process.

We look forward to receiving your revised manuscript.

Kind regards,

Li Zeng

Academic Editor

PLOS ONE

Journal Requirements:

Reviewers' comments:

Reviewer's Responses to Questions

**Comments to the Author**

1. If the authors have adequately addressed your comments raised in a previous round of review and you feel that this manuscript is now acceptable for publication, you may indicate that here to bypass the “Comments to the Author” section, enter your conflict of interest statement in the “Confidential to Editor” section, and submit your "Accept" recommendation.

Reviewer #1: (No Response)

2. Is the manuscript technically sound, and do the data support the conclusions?

Reviewer #1: Yes

3. Has the statistical analysis been performed appropriately and rigorously? 

Reviewer #1: I Don't Know

4. Have the authors made all data underlying the findings in their manuscript fully available?

Reviewer #1: No

5. Is the manuscript presented in an intelligible fashion and written in standard English?

Reviewer #1: Yes

6. Review Comments to the Author

Reviewer #1: This manuscript by Choa Ji et al describes a compressed sensing technique for fluorescence lifetime imaging. This is the second time I have reviewed this paper after major revision. This manuscript is better written and much more understandable.

I appreciate the extra background information and detail provided to put the proposed algorithm in context. There has also been significant revision of the methods section and the addition of new experimental results. The authors chose to include a new experiment with a different substance. It would have been better to use a sample that has at least two different regions, containing different substances with different lifetimes, as I suggested in the first review.

There are minor additions still required before I can recommend publication.

The ground truth for the simulated images are still not included. Figures 4 and 5 show the results of 6 algorithms, alongside those results I need to see the ground truth images from the original dataset.

The test data sets are "runner" and "drop". Please describe how these were acquired. What hardware was used etc.

Abstract: By "high frames", do you mean "high frame rates"?

Abstract: What does "large-scale" mean? Please clarify in the manuscript.

Abstract: It is not clear what "runner test sets" means, please clarify or replace with "test data sets".

Figure 1: Please describe the acronyms (M1 etc) in the figure legend.

Line 95: The purpose of M1 is not clear. Is it crucial to the algorithm or just to provide some arbitrary spatial structure?

Line 110: Remove "In the CUP system" and start the sentence with "A DMD is..."

Line 112: Please clarify the sentence, "The binary coding is isolated..."

Line 141: Please clarify what is meant by "continuous spatial feathers".

Line 148: "We can Assume" should be "We define"

Line 237: Please define PSNR and SSIM.

7. PLOS authors have the option to publish the peer review history of their article (what does this mean?). If published, this will include your full peer review and any attached files.

Reviewer #1: No

---

## [Author Response · Author response to Decision Letter 1]

24 Jun 2022

Replies to Journal

Comment 1: Please review your reference list to ensure that it is complete and correct. If you have cited papers that have been retracted, please include the rationale for doing so in the manuscript text, or remove these references and replace them with relevant current references.

Answer:

We have checked the reference list and confirmed that it is complete and correct, and there is no retracted article.

Replies to Reviewer #1

Comment 1: The ground truth for the simulated images are still not included. Figures 4 and 5 show the results of 6 algorithms, alongside those results I need to see the ground truth images from the original dataset.

The test data sets are "runner" and "drop". Please describe how these were acquired. What hardware was used etc.

Answer: 

We added the original images labeled ‘Ground Truth’ in Figures 4 and 5, and we added the test data sets acquisition links in 219 lines of the manuscript. Moreover, in 262 lines of the manuscript, we describe the system environment for algorithm execution

Comment 2: Abstract: By "high frames", do you mean "high frame rates"?

Answer:

The "high frames" is intended to express the number of imaging frames. But the "high frames" is an inappropriate description. We have changed it to the “deep frame sequences”.

Comment 3: Abstract: What does "large-scale" mean? Please clarify in the manuscript. 

Answer:

For the problem of image compression and reconstruction, the “large-scale” generally refers to the deep compressed frame sequences. For example, the number of compressed images exceeds 30.

The term “large-scale” can be found in reference “Plug-and-Play Algorithms for Large-Scale Snapshot Compressive Imaging”. For a more reasonable description, we change the “large-scale FLIM” to the “large-scale FLIM problem” in the manuscript.

Comment 4: Abstract: It is not clear what "runner test sets" means, please clarify or replace with "test data sets".

Answer:

We changed the "runner test sets" with the "test data sets".

Comment 5: 

Figure 1: Please describe the acronyms (M1 etc) in the figure legend.

Line 95: The purpose of M1 is not clear. Is it crucial to the algorithm or just to provide some arbitrary spatial structure?

Answer:

We added the acronyms description in the figure1 legend. 

M1 is only to highlight the 2D spatial distribution and has nothing to do with the algorithms.

Comment 6: 

Line 110: Remove "In the CUP system" and start the sentence with "A DMD is..."

Answer:

We removed the "In the CUP system" and modified our grammar mistakes.

Comment 7: 

Line 112: Please clarify the sentence, "The binary coding is isolated..."

Answer:

We changed the original sentence to " We randomly generated multiple groups of coding layouts through MATLAB, and selected the best coding layout by simulation results "

Comment 8: 

Line 141: Please clarify what is meant by "continuous spatial feathers".

Answer:

The "continuous spatial feathers" is intended to express the smoothing properties of natural signals. We replaced the "continuous spatial feathers" with the “spatial smoothing properties”

Comment 9: 

Line 148: "We can Assume" should be "We define"

Answer:

We have replaced the "We can Assume" with the "We define".

Comment 10: 

Line 237: Please define PSNR and SSIM.

Answer:

We added the formulas and descriptions of the PSNR and SSIM at line 238 of the manuscript

---

## [Decision Letter · Decision Letter 2]

1 Jul 2022

Compressed fluorescence lifetime imaging via combined TV-based and Deep Priors

PONE-D-22-04257R2

Dear Dr. jinshou,

We’re pleased to inform you that your manuscript has been judged scientifically suitable for publication and will be formally accepted for publication once it meets all outstanding technical requirements.

Kind regards,

Li Zeng

Academic Editor

PLOS ONE

Additional Editor Comments (optional):

Reviewers' comments:

Reviewer's Responses to Questions

**Comments to the Author**

1. If the authors have adequately addressed your comments raised in a previous round of review and you feel that this manuscript is now acceptable for publication, you may indicate that here to bypass the “Comments to the Author” section, enter your conflict of interest statement in the “Confidential to Editor” section, and submit your "Accept" recommendation.

Reviewer #1: All comments have been addressed

2. Is the manuscript technically sound, and do the data support the conclusions?

Reviewer #1: Yes

3. Has the statistical analysis been performed appropriately and rigorously? 

Reviewer #1: Yes

4. Have the authors made all data underlying the findings in their manuscript fully available?

Reviewer #1: Yes

5. Is the manuscript presented in an intelligible fashion and written in standard English?

Reviewer #1: Yes

6. Review Comments to the Author

Reviewer #1: Thank you for your revisions. I now recommend publication.

Please consider making your code available according to the policy: https://journals.plos.org/plosone/s/materials-software-and-code-sharing

7. PLOS authors have the option to publish the peer review history of their article (what does this mean?). If published, this will include your full peer review and any attached files.

Reviewer #1: No

---

## [Editor Report · Acceptance letter]

3 Aug 2022

PONE-D-22-04257R2 

Compressed fluorescence lifetime imaging via combined TV-based and Deep Priors 

Dear Dr. Tian:

I'm pleased to inform you that your manuscript has been deemed suitable for publication in PLOS ONE. Congratulations! Your manuscript is now with our production department. 

Kind regards, 

on behalf of

Professor Li Zeng 

Academic Editor

PLOS ONE